# The Prevalence of Teleophthalmology in the Piedmont Region of Italy: Current Situation and Future Perspectives

**DOI:** 10.3390/ijerph19148608

**Published:** 2022-07-15

**Authors:** Raffaele Nuzzi, Floriana Deraco, Simona Scalabrin

**Affiliations:** Eye Clinic, Department of Surgical Sciences, University of Turin, 10126 Turin, Italy; florianaderaco@gmail.com (F.D.); simona.scalabrin@unito.it (S.S.)

**Keywords:** telemedicine, teleophthalmology, piedmont, teleophthalmology Italy

## Abstract

Purpose: This cross-sectional study aims to evaluate the actual prevalence of telemedicine, with a particular attention to teleophthalmology programs, in the Piedmont region of Northern Italy and investigate the prospects of the discipline, comparing the situation with the rest of Italy. Information about the current teleophthalmological development is important to assess the state-of-the-art of innovation in medicine in different areas of the world so that there can be a healthy comparison and evaluation of progress and backlog. Methods: We sent questionnaires to every Local Healthcare Authority and gathered the answers in five distinct categories. Due to the COVID-19 pandemic, we collected information by telephone. We investigated five primary areas: what type of telemedicine services are currently in operation, how they are managed, the presence of any evaluation of patient satisfaction and cost-savings, and the prospects of future teleophthalmology programs to be implemented. Results: Only 2% of the total telemedicine programs are in the field of ophthalmology. The greater parts are in the field of cardiology (15%) and endocrinology (13%). Currently, only one program of teleophthalmology exists in the territory, and at least four more projects are waiting to be approved or funded. Surveys about patient satisfaction were not provided and there was no evidence of cost-saving. Conclusions: Teleophthalmology in Piedmont is slowly developing, although there is a lack of a unified network for storing and managing patients’ data. During the COVID-19 pandemic, telemedicine usage drastically increased, and there is a need to evaluate this trend to understand its place in the future of medicine.

## 1. Introduction

Telemedicine is a discipline that enables the delivery of public health services at a distance. It uses a reliable and protected transmission of information in the form of text, sound, image, and video. Its usefulness is more pronounced in rural areas or developing countries, where there is a lack of adequate medical infrastructure, or during outbreaks of infectious diseases, pandemics, and natural disasters. For this reason, it can play a key role in global health development. The technologies used are defined as “Information and Communication Technologies” (ICT, the set of methods and technologies that implement the systems of transmission, reception, and processing of information), provided that they are managed and supervised by the NHS or delegated bodies [1,2].

In addition, there is the possibility for professionals from all over the world to connect to discuss a clinical case together and increase the quality of a diagnosis. Telemedicine can also integrate into preventive medicine by allowing patients to access information regarding their health and for monitoring chronic conditions [1].

Telemedicine services should be assimilated into any diagnostic/therapeutic health service, even if they are not replaceable to it as far as the quality of the personal doctor–patient relationship is concerned. This service should be considered more as an integration that can improve the effectiveness, efficiency, and appropriateness of normal health services. One of the advantages is the possibility of telematic patient monitoring, avoiding long waiting lists, and reducing hospitalization [2].

In 2019, the WHO began developing a framework for the adoption of digital innovations in healthcare, with recommendations based on benefits, risks, acceptability, feasibility, resource use, and equality considerations, to improve healthcare coverage and sustainability throughout the world. There were five priorities identified in the use of telemedicine, as: complementary to health services, the transmission of personalized information, support of health decisions made by health professionals, tracking of digital health information, and education.

Telemedicine can facilitate more equitable and efficient distribution of the limited healthcare resources to which we have access. In addition, it can provide services in remote areas where there is a lack of healthcare professionals, reduce travel and therefore the pollution associated with it, connect patients with rare diseases to specialists, and enable the comfort and safety of weaker patients who would be at risk during travel. Telemedicine can reduce waiting times, especially in acute disease settings where a local urgent care center is not immediately available [3]. In recent years, thanks to the availability of more effective data compression techniques and faster networks, it has become possible to exchange data of increased size. It also became possible to utilize the smartphone as a tool for e-health, allowing the patient to follow-up on their health through mobile applications precisely created to monitor parameters and to educate themselves through the process [4].

In ophthalmology, examples of telemedicine devices include ForeseeHome™ (Notal Vision, Inc., Tel Aviv-Yafo, Israel) and PsyPad™ (University of Melbourne, Parkville, VIC, Australia), visors connected to online platforms, aimed at monitoring the progression of degenerative maculopathy in chronic patients and alerting them if it detects a scotoma or a viewing defect. The technology Alleye™ (Oculocare Medical Inc., Zürich, Switzerland) for smartphones evaluates the loss of vision, normally non-perceptible by the patient, in edemigenous pathologies and macular degeneration [3].

The advantage presented by these innovative technologies is evident, especially when considering the possibility for patients who do not need hospitalization of avoiding a journey to the hospital. It also presents a way of avoiding nosocomial infections and contagions during a pandemic. Patients can receive medical help from a remote doctor from the comfort of their own home, which is particularly important when the elderly or disabled are involved [4].

### The Situation in the Piedmont Region

Regarding the Piedmont region of Northern Italy, a document dated 2018 explored the presence of telemedicine in the regional area, through questionnaires sent to all Regional Health Authorities (ASLs). The census covered 45 projects and was completed in June 2017.

Most of the services surveyed (15) were telemedicine (teleconsultation, telehealth, telehealth cooperation), 10 were telemonitoring, 11 were a combination of telemedicine and telemonitoring services, and there were 9 other types of services (telereferral, telehealth, and telerehabilitation). Telereferral had the highest number of patients served (approximately 465). Cardiology and diabetology were the areas with the most available telemedicine services, particularly telemonitoring for cardiology.

Funds to finance these projects were both public (38%) and private (40%). Benefits of establishing services included: raising the level of patient quality of life (37% of projects), improving appropriateness and timeliness of care (30%), and strengthening the ability of healthcare professionals to operate (26%).

Among the weaknesses found were (40% of respondents): lack of recognition of a fee for telemedicine services, lack of technological components, insufficient coverage of the telematics network in some areas (28%), and difficulties in the regulatory framework of the professional figures involved in the provision of the service (26%).

Subsequently, in 2020, the Resolution of the Regional Council (DGR) No. 6-1613/2020 of 3 July 2020 was issued regarding telemedicine services, aimed at public health agencies, and accredited and contracted private individuals. This was helped by the pressure of the COVID-19 emergency, during which in-person health services slowed down considerably, intending to establish a structure of stable rules that could be used even after the pandemic was over.

According to the National Plan for Chronicity (agreement sanctioned under art. 4 of Legislative Decree 28.8.1987, No. 281, on 15 September 2016—Rep. atti160/CSR), the care of the chronic patient can be favored significantly at home by digital health technologies (the so-called e-Health), especially telemedicine and telecare. The DGR aimed to dictate a set of rules to guide the application of telemedicine. Some of the conditions to be admitted into a telemedicine program include: being in a state of known chronic illness in need of follow-up for a long period, being included in a PDTA (diagnostic–therapeutic pathway of care), needing a consultation to modify or adjust therapy or to renew a therapy plan, and needing a consultation about the results of diagnostic or staging examinations performed.

Finally, it was stated that it is not advisable to offer telemedicine services to patients with acute pathologies or in need of intensive hospital care [5].

This study aims to investigate the usage of telemedicine, provided by public health services, in the Piedmont territory, with a particular attention to the current prevalence of teleophthalmology and its prospects, also evaluating the programs that are not yet in place, but still waiting for approval or funding.

## 2. Materials and Methods

We conducted a cross-sectional study during the last months of 2021.

We sent a questionnaire to 12 ASLs of the Piedmont region, 4 of which are located in the metropolitan area of Turin.

The ASL is a public body belonging to the Italian public administration, the purpose of which is to provide health services in each territorial area, which may be a municipality, a province, or a collection of cities. Each Italian region is divided into several ASLs, while the Piedmont region is divided into 12. Every ASL caters to a population that can differ in quantity and quality (age, race, economic status, etc.) since the territory has both cities (the biggest one is Turin, with almost 900,000 inhabitants), countryside, and mountains. The total population of Piedmont is on average 4,400,000 people.

We included all 12 ASLs in the study, but we did not receive a response from 1 of them.

For each ASL, the professional figure in charge of telemedicine was addressed each time.

Due to the COVID-19 pandemic, we collected the answers telephonically, with a discussion-based approach (open-ended questions) instead of a closed-answer one, and the results were assigned to five major categories:Which telemedicine services were active in the area at the time?The details of how they were implemented.How patients perceived them.The impact they had in economic terms.What were the prospects regarding the implementation of teleophthalmology services?

We created the questionnaire investigating the current use of telemedicine in general and the development of teleophthalmology in particular. We utilized a descriptive statistical evaluation of the outcomes.

To understand the current general usage of telemedicine, we considered all the specialty practices offered in the territory and asked each ASL if services for a specific specialty were also offered through telemedicine. In this paper, we paid particular attention to teleophthalmology services.

## 3. Results

Of 45 total programs, cardiology has the highest number of absolute frequency (7), followed by endocrinology (6) and psychology (4). The specialties that have two positive answers are rehabilitation/speech therapy, pneumology, psychiatry, vulnology, nephrology, neurology, pediatrics, and gynecology. Lastly, telemedicine specialties only present in one ASL out of the total are ophthalmology, dermatology, geriatrics, surgery, radiology, transfusion medicine, audiology, long-time hospitalization, COVID-19, oncology, gastroenterology, and infective medicine.

Most of the telemedicine programs offered were in the departments of cardiology (15%) and endocrinology (13%). It is possible to see the distribution in Figure 1.

Currently, only one program of teleophthalmology is offered in the regional territory. It consists of a remote assessment for diabetic retinopathy, with the patient still present in the facility.

Surveys regarding the satisfaction of the patients involved in the telemedicine programs were not provided, but a positive outcome was verbally reported by a good portion of the ASLs.

There was no information available regarding the economical aspect, but internal audits are planned in the future.

We then focused on prospects regarding the implementation of teleophthalmology services. Four teleophthalmological projects are waiting to be implemented (see Table 1), which include:A service for the maculopathy’s ambulatory of the Ospedale Oftalmico of Turin, in which patients treated with intravitreal injections will be evaluated by ophthalmologists through telemedicine and only sent to the hospital if further treatment is required.A project called “Retina Online”, in which patients needing a retinography will be checked with the use of a transportable instrument. The examination will be conducted with the coordination of territory health services and performed by an ophthalmologist or another suitably trained practitioner (e.g., orthoptist or nurse), without mydriasis. The images acquired in digital format will be sent to the referring ophthalmologist’s station via the company intranet or an alternative IT support. As this is a completely new field from the practical point of view of manual training, it is supposed to start with a timeframe foreseen for an initial trial period, with an approximate time of 15 min for the execution and about 15 min for the report. The report will be available for collection in a practical manner still to be established. A large catchment area (in the order of 1000) is assumed, regarding diabetic patients, a proportion of hypertensive patients, as well as those with medical conditions, chorioretinal diseases (e.g., high myopia, trauma outcomes, choroidal nevi, maculopathies, etc.), and glaucoma (monitoring of the optic papilla with stereophotography).A screening for diabetic retinopathy with remote reporting after evaluation with artificial intelligence. It is expected to be performed using non-mydriatic digital retinography by nurses belonging to the Diabetes Unit and/or the District Units. The retinography will then be uploaded by the personnel who carry them out onto the Smart Digital Clinic digital folder already in use by the ASL. The images will be consulted remotely by a single reading center made up of ophthalmology specialists. The referring doctors will work with an additional company fund. The images may be “pre-referenced” by artificial intelligence software, but they must always be re-evaluated by a referring doctor from the reading center. It is expected to include 2500 screened patients per year at the company level, at a total cost of EUR 9000 (plus the purchase of a non-mydriatic digital retinograph with true confocal technology of EUR 12,000 + VAT).At-home monitoring in glaucomatous patients with a low risk of progression. It will be offered for patients in follow-up at the outpatient clinic and clinically known to the specialist. It requires that, after evaluation of the risk of disease progression (IDR), they will be able to have a glaucoma eye examination in a televised mode (a maximum of one televised visit per year, the other examinations and/or services must be carried out in the facility) using a home-use tonometer that will be provided to them for 15 days. During the at-home monitoring period, several measurements will be taken at separate times during the day to draw up a tonometry curve. The method used will be rebound tonometry with disposable probes. The measurement data will be saved on an internal memory of the instrument with the possibility of downloading them later and printing a report for the patient. The activity is foreseen to cover about 40 patients. The expected costs are the purchase of two I-care home tonometers for a total cost of EUR 5000 + VAT.

## 4. Discussion

Information about the current development of telemedicine is important to assess the state-of-the-art of innovation in different areas of the world so that there can be a healthy comparison and evaluation of progress and backlog.

The last document that explored telemedicine in general in the Piedmont area dates back to 2017 [5] and does not have a particular focus on teleophthalmology, nor does it explore the programs waiting for approval.

Such research does not exist in the literature, and it is an underdeveloped and under-considered area, hence our interest to contribute to bringing its current situation to light.

Considering our results, we found that the most common specialties to have a telemedicine program associated with them are cardiology and endocrinology. This result can be explained by the abundant presence of devices to monitor parameters in chronic patients, especially those affected by heart failure and diabetes. For chronic heart failure, the cardiac implantable electronic devices used to treat bradyarrhythmia and tachyarrhythmia nowadays have been implemented with the feature of remote monitoring, to gather, store, and transmit data regarding the patients’ parameters to clinicians [6].

Regarding diabetology, telemedicine has been shown to improve the outcomes of patients, using glucometers that store and forward the parameters of glycemia, hematocrit, hemoglobin, and ketones, and through video, check-ups to discuss dieting and exercising [7].

Financing for all telemedicine programs came both from private sources and the National Sanitary System. There was no evidence of any findings of cost-saving related to the application of telemedicine, but there could be in the future.

The platforms used to store and forward patients’ data were always different. Mostly, each ASL uses its company platform, but often they also use platforms created especially for a device by the producing company.

### 4.1. Economical Aspects of Telemedicine

Regarding the economical aspect, the Piedmont region in 2020 has decided that the same means of payment and the same costs have to be applied to telemedicine ambulatory services, including the possible payment of the ticket—a fee paid by the citizen as a contribution to the sanitary expense [8].

There still have not been studies to assess the economic impact of telemedicine on the sanitary budget, and only one of the ASLs has a plan for an internal review soon.

Moreover, an eventual assessment of the economic impact of telemedicine would not be straightforward. It is important to note that for the patient, using fewer travel resources (fuel, vehicle use) and the possibility of not losing working hours contribute to the overall savings. For the clinician, visit times are reduced, resulting in better allocation of specialist doctors’ time, and better management of medical and drug prescriptions.

### 4.2. The State-of-the-Art of Teleophthalmology

Only 1 out of the total 45 telemedicine programs is in the field of ophthalmology. In the future, this number may well expand if the suggested programs are approved and instated.

There are very few examples of a successful teleophthalmology program set in Italy in international literature.

Among these, we have found the experience of Ospedale Maggiore of Bologna, which through teleophthalmology provided continuity of care and protected the health of both health workers and patients by avoiding direct contact during the height of the COVID-19 pandemic. To instruct patients and their caregivers on how to approach the technologies, they have produced tutorials to show how to conduct the video consultation. For the interaction with the patient, they have used a telematic platform, with both a “desktop” version and a “mobile” version, where they could select various tests and collect data. The most appropriate clinical and therapeutic decision was then made by evaluating the data and results on the platform. Some of the major applications have been in pediatric ophthalmology (e.g., visual rehabilitation after occlusive therapy in amblyopia), medical retina for the follow-up of exudative maculopathies, and after the intravitreal treatment cycle to monitor the clinical stability. After one year, patients expressed a high level of satisfaction, particularly those who were chronically ill, frail, disabled, and coming from more peripheral districts. Moreover, they received satisfied feedback from family members and caregivers, who had less responsibility and gained in free time [9].

In Lazio region of Central Italy, there is evidence of another program, where a triage service for ophthalmological emergencies was offered, which could be managed entirely by telemedicine or sent to specialist centers. During the experience, they reported mostly elderly users, and especially for superficial eye diseases [10].

In the ophthalmology department of Rovereto (Trento), one program began during the pandemic, aimed mainly at the pediatric area, using a smartphone application called “TreC Oculistica”. The program focuses on screening and follow-up, while first visits accounted for about 3% of the total number of services provided [11].

There is also a service called “Vista in salute” which consists of an extraordinary mobile screening project funded by the 2019 Stability Law and extended until 2023. The project is conducted by the IAPB, the Italian branch of the International Agency for the Prevention of Blindness. Its aim is the early diagnosis of retinal pathologies and the dissemination of information about visual impairment.

A large mobile clinic (mounted on a truck) moves from square to square in the main Italian cities and provides free examinations at several locations, reserved for people over 40 years old.

OCT examinations and photography of the ocular fundus, autorefractometry, and tonometry are offered, and the main pathologies sought are glaucoma, maculopathies, and diabetic retinopathy. The project also aims to collect epidemiological data that will be used to set up a national database to assess the impact of eye diseases and develop public health policies for visual protection [12].

### 4.3. Prospects of Teleophthalmology

According to the IAPB, teleophthalmology is not able to replace eye examinations because of the lack of empathy between the doctor and the patient and the lack of patients’ choice of doctor. Instead, its role should be to support eye examination because of its speed and the possibility of making healthcare more equitable [13].

One of the reasons why teleophthalmology has not caught on as well in Italy as in other parts of the world could be found in the lack of predisposition by ophthalmology specialists in the country. In a recent questionnaire submitted by the Italian Society of Forensic Ophthalmology (SIOL) in 2021 to assess the use of teleophthalmology in Italy, we can see the percentages of interest in the subject. The responses came from around 400 ophthalmologists, mostly from the north (55%), then the center (26%) and the south (19%).

Concerning the question that asked for an opinion on the capacity of telemedicine to provide health services at a distance, about 55% believe that it is currently premature, 25% believe it is already available, and 20% identify it as the future of the medical practice, 25% believe that e-Health services are premature in the field of ophthalmology, 60% of respondents stated that they have not yet had any real “direct contact” with telemedicine, and 40% believe that a doctor–patient relationship conducted via telemedicine may affect the relationship of trust.

However, over 80% of respondents, although convinced that a complete eye examination cannot be completely avoided, believe that telemedicine can be used for remote reporting of instrumental examinations if they are conducted by experienced staff such as Orthoptists Assistants in ophthalmology.

From what emerged from the study, it appears that only a small percentage of Italian ophthalmologists have dealt with teleophthalmology (with higher percentages of young doctors), and that a large proportion of them are reluctant to do so [14].

In Italy, there is a lack of a unified network that brings together the services of teleophthalmology at a national level. Currently, each department chooses the platform most congenial to the needs of its patients and employees and it is therefore very difficult to nationally collect information and standardize teleophthalmology services to reach a common standard.

Thanks to our study and the collection of data from all the regional health entities of Piedmont, it emerged that each ASL used its system to store and make data available. The platforms used in a lot of cases were those provided by the pharmaceutical companies producing the devices used to track parameters (e.g., glucometer, mechanical ventilator). Many times, the platforms used for the video calls were the external and most commonly used ones, such as Skype, WhatsApp, and Cisco Webex.

A few of them forwarded patients’ information from their internal database onto the regional Electronic Health Record. The Electronic Health Record (Fascicolo Sanitario Elettronico) is an online collection of a citizen’s health records, including their vaccine history, instrumental examinations, dematerialized prescriptions, future health appointments, and screenings. It is effectively a tool to track and consult their entire health life history, sharing it with health professionals to ensure a more effective and efficient service. There is the possibility of interoperability among the regions of Italy, that allows searching, retrieval, registration, deletion of documents, and index transfer of the Electronic Health Record [15]. Each region manages its own Electronic Health Record, and access is possible through SPID, or “Sistema Pubblico di Identità Digitale” (Public Digital Identity System), a service that permits access to the online services of the Public Administration [16].

At the European level, the NIS (Network and Information Security) Directive was approved in 2016, which is the basis for the possibility of standardization of the control of transmission and acquisition of sensitive data between different states through the adoption of common security measures [4]. Currently, there are two projects under development that will allow cross-border healthcare between the EU Member States through digital means, as the European Commission is working on setting up an IT network to ensure the interoperability of e-Health services through the Connecting Europe Facility (CEF) European program. Thanks to the program, it will be possible for all European citizens to use the documents contained in the Electronic Health Record in whichever Member State they are in, facilitating and improving the European health service for citizens [15].

In Italy, as in the rest of the world, the patients who suffered most from the interruptions in specialist activities during the pandemic were those suffering from chronic degenerative retinal diseases. This context was propitious for the inclusion of teleophthalmology in the treatment and diagnosis of these and other fragile patients, giving rise to the possibility of a new digital health revolution, with major social and health implications.

Compared to the Italian context, Piedmont is not lagging. Public evidence of successfully established teleophthalmology programs is few and concentrated in the larger cities. In addition, there is a lack of evidence from Southern Italy. According to the SIOL 2021 survey, most ophthalmology specialists in Italy are wary of digital healthcare and a low percentage have experienced it.

The areas most in need of teleophthalmology are easily the rural, country, and mountain areas, where the inhabitants are more isolated and far from the nearest ophthalmology facility. Possible applications are both in the emergency field, reducing the probability of permanent damage to eyesight, with the possibility of remote consultation within a brief time, and in the case of long-term follow-ups of chronic diseases, such as degenerative maculopathy and diabetic retinopathy. The type of patients suffering from such diseases are usually the elderly, who would therefore gain several advantages from being able to make visits from home or the ASL facilities in their area, to avoid the stress of long journeys and exposure to pathogens in hospitals. On the other hand, these types of patient are often unfamiliar with the technology required for a quality video consultation. It is therefore important that the focus is on the assistance and education of both patients and their caregivers. Furthermore, it is important to provide the appropriate equipment, also with the assistance of trained staff from local health authorities and pharmacies.

It is also necessary to implement a national uniform teleophthalmology network, both because of the need to protect sensitive patient data and to achieve a common standard of excellence in services and to collect data in a national database that can develop epidemiological evidence. The first step is to improve the Electronic Health Record and make it a standard to upload all the telemedicine-gathered data on it. The interconnectivity of all the regions’ Electronic Health Records allows for the basis to form a national telemedicine network, which in the future will also be able to expand to all of Europe if the latest projects prove effective.

Teleophthalmology can facilitate more equitable and efficient distribution of the limited health resources we have access to, reducing the need for an in-person consultation, and allowing the collection of high-quality and structured data that can be used for personalized management.

### 4.4. Limitations and Strengths of the Study

Among the limitations, our questionnaire was not validated nor standard, so it is not possible to objectively compare all our data with other studies. Furthermore, since the questions were answered orally, there could have been a freer interpretation of the answers on our part. Our approach was based on discussion and qualitative findings and lacked objectivity and systematicity.

However, this study has a potentially important role in the development and spreading of teleophthalmology globally, providing examples from a real-life setting and documenting the growth of the discipline in time. It shows how a region can shift the resources to better suit a different condition of society, such as a pandemic. One of the strengths of this study is that we have interacted with all the different professional figures in charge of telemedicine in one of the most developed regions of Italy, covering all the regional areas. We could better understand the context of the ASL and combine that with the answers to have a wider overview of the current and future situation.

Further research is required to monitor the development of telemedicine in the future and the impact it will have both on the health of citizens and on the economy, as well as on the work environment of medical professionals.

## 5. Conclusions

The COVID-19 pandemic has stimulated and accelerated the digital health revolution, a technology that is capable of renewing and expanding the activities and offerings of the National Health System, keeping it up to date with the rest of the world and to the needs of society in a constant state of transformation.

We can divide the current situation in Piedmont into two different fields:As regards telemedicine in general, we found excellent applications, both for telemonitoring, especially of diabetic patients or those with heart failure, conducted utilizing devices at home, and for the follow-up of chronic patients through video consultation. The latter solution was particularly popular during the period of the COVID-19 pandemic, making it possible to reduce patient traffic within healthcare facilities. However, it has the potential to become an ongoing service offering advantages to both the patient and the clinician in terms of savings and convenience.Regarding teleophthalmology, on the other hand, its application is currently very scarce, almost non-existent, with only one existing program at ASL TO5. Under the pressure of the pandemic, however, we are seeing at least four new projects being developed that have particularly good prospects for soon becoming stable ophthalmology patient management programs. Under the impetus of these new projects, new ones may be created, using the footprint of those already in place as a guide, to spread teleophthalmology to the whole of Piedmont.

## Figures and Tables

**Figure 1 ijerph-19-08608-f001:**
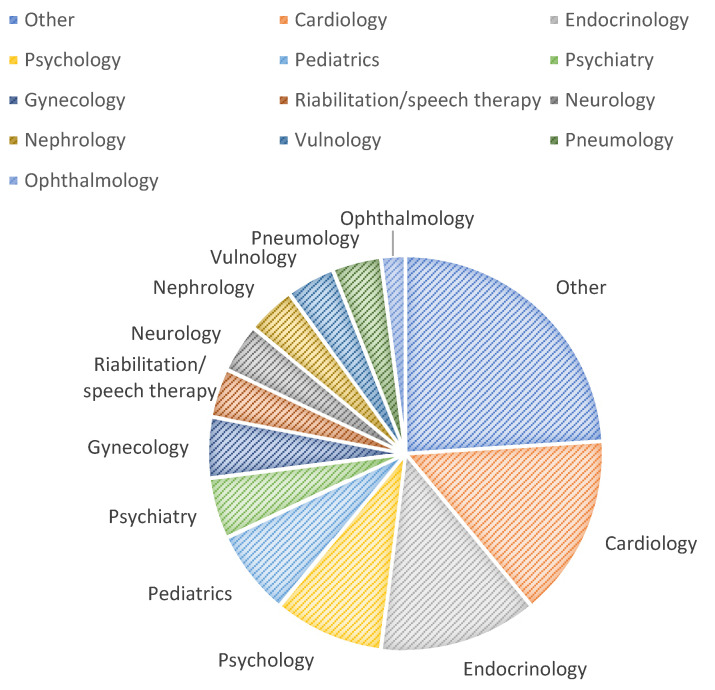
Distribution of telemedicine programs in the Piedmont territory.

**Table 1 ijerph-19-08608-t001:** Implemented teleophthalmology programs and programs waiting for approval and/or funds.

Status	Program	Territory	Aim
Implemented	Assessment of diabetic retinopathy	ASL TO5	Remote assessment of diabetic retinopathy, for patients still present in the hospital facility.
Waiting for approval/funds	Maculopathy’s ambulatory	ASL Città di Torino (TO1, TO2)	Follow-up of patients treated with intravitreal injections through telemedicine and in-person only if further treatment is required.
Waiting for approval/funds	Retina Online	ASL TO3	Retinography collection with the use of a transportable instrument, with the coordination of territory health services and performed without mydriasis.
Waiting for approval/funds	Diabetic retinopathy screening	ASL TO5	Screening for diabetic retinopathy with remote reporting after evaluation with artificial intelligence.
Waiting for approval/funds	Home monitoring for glaucoma	ASL TO5	Home monitoring in glaucomatous patients with a low risk of progression, using a home-use tonometer to draw up a tonometric curve.

## Data Availability

Not applicable.

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
