# Peer review of "The Prevalence of Teleophthalmology in the Piedmont Region of Italy: Current Situation and Future Perspectives"

_ijerph, 2022, doi:10.3390/ijerph19148608_

Round 1
Reviewer 1 Report
This study aims to evaluate the prevalence of teleophthalmology programs in the Piedmont region of Italy and the future prospects of the discipline, compared to the rest of Italy. The paper cannot be accepted in its present form as it needs further improvements.
-I would encourage the authors to present a more compelling argument regarding the originality and relevance of their work relative to the previous research work in a clear, easy-to-understand, and verifiable fashion.
- Abstract: The text must be carefully revised. Some sentences contain mistakes. Avoid using the acronym. Explain it in detail. Discuss what the Research Gaps/Contributions are?
- The English writing of the paper is required to be revisited. Please check the manuscript carefully for typos and grammatical errors. Avoiding split infinitives can help your writing sound more formal.
-Conclusions should be more concrete. They should be summarized in 3-5 bullet points that clearly show the conclusions of this study. In addition, since it is a review, it is essential to indicate future lines of research.
-Rather than a research article, it looks like a normal report. The authors need to rewrite the paper or reconsider the research content before being considered for publication in this journal.
Author Response
Dear reviewer, we thank you for your suggested corrections. We find now that the work has been highly implemented by them.
1) We have added reasons why our work is original and relevant. Such research does not exist in the literature.
2)We have revised the abstract and added contributions and limitations.
3) We revised the language and modified sentences that were not structured correctly.
4)We shortened the conclusions to make them more concise.
5)We hope now that the article is shaped more appropriately.
Thank you, best regards.
Reviewer 2 Report
This is an interesting study on health care during COVID-19.
Please consider the following changes to improve the scientific soundness of the manuscript.
1. The Abstract section should be more informative. Please add some data in the Results section (% values or major findings etc.)
2. The citation method is unclear [1, pp. 197,198] what does it mean? Please use an official citation recommended by the MDPI journals. If the author wants to indicate pages, they can do it in the te Reference section in line with citation guidelines when citing books.
3. Please clearly define the study aim.
4. Methods section is too short. Please clearly define the methodology. Type of study (cross-sectional ?), eligibility criteria, measures and outcomes, study design and populations as well as statistical methods.
5. The results section covers citations. If the Authors want to publish this paper as an original article, the results section should be revised to match the guidelines for the original papers
6. Please add one paragraph on limitations of this study
7. Conclusions are too extensive. Please provide short and informative conclusions. The rest of the text may be moved to the discussion
Author Response
Dear reviewer, thank you for your suggested corrections. They were constructive and useful.
1) We added more information in the abstract.
2)We corrected the citation method, following the guidelines.
3)We stated the study aim, to make it more clear.
4)Methods were extended, to better explain our process.
5)We moved the citations from the results section to the discussion.
6)Limitations and strengths were added.
7)We shortened conclusions and summarized them in bullet points.
Thank you very much for your contribution,
best regards
Round 2
Reviewer 1 Report
The paper has been improved and it can be forwarded for publication.
Reviewer 2 Report
The manuscript was significantly improved. The current version is scientifically sound and provides all necessary data.